Public domain. CC0 1.0.



# Brief Communication: Evaluating Snow Depth Measurements from Ground-Penetrating Radar and Airborne Lidar in Boreal Forest and Tundra Environments during the NASA SnowEx 2023 Campaign

Kajsa Holland-Goon[1], Randall Bonnell[2,1], Daniel McGrath[1], W. Brad Baxter[3], Tate Meehan[4], Ryan Webb[5], Chris Larsen[6], Hans-Peter Marshall[7], Megan Mason[8,9], Carrie Vuyovich[8]

[1]Department of Geosciences, Colorado State University, Fort Collins, Colorado, USA
[2] U.S. Geological Survey, Water Resources Mission Area, Denver, Colorado, USA
[3]Cold Regions Research and Engineering Laboratory, U.S. Army Corps of Engineers, Fairbanks, Alaska, USA
[4]Cold Regions Research and Engineering Laboratory, U.S. Army Corps of Engineers, Hanover, New Hampshire, USA
[5]Department of Civil and Architectural Engineering & Construction Management, University of Wyoming, Laramie, Wyoming, USA
[6]Geophysical Institute, University of Alaska, Fairbanks, Alaska, USA
[7]Department of Geosciences, Boise State University, Boise, Idaho, USA
[8]Hydrological Sciences Laboratory, NASA Goddard Space Flight Center, Greenbelt, Maryland, USA
[9]Science Systems Applications Inc., Lanham, Maryland, USA

*Correspondence to*: Kajsa Holland-Goon (krhollgoon@gmail.com), Daniel McGrath (daniel.mcgrath@colostate.edu)

Public domain. CC0 1.0.

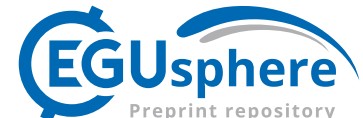

**Abstract.** We evaluated ground-penetrating radar (GPR) and airborne lidar retrievals of snow depth collected during the NASA SnowEx 2023 campaign in Alaskan tundra and boreal forest environments along 44 short (3–12 m) transects. Compared to in situ observations, we identified modest biases for GPR snow depths (bias <0.03 m in tundra, +0.06 m in boreal forests) and larger biases for lidar snow depths (bias +0.19 m at a tundra site, –0.16 m in boreal forests) related sub-snow vegetation, tussocks, and seasonally dynamic ground. These complex surface environments present a challenge to established methods, which needs to be considered when evaluating novel remote sensing approaches.

## 1 Introduction

In high-latitude (>60°) terrestrial systems, snow plays a crucial role in the hydrologic cycle and modulates the surface energy balance. Snow also strongly impacts the ecology of these regions: snow influences caribou winter range selection (Pedersen et al., 2021) and vegetation phenology (Kelsey et al., 2021), and provides winter refuges for a diverse range of animals (Penczykowski et al., 2017). Snow cover extent in the Arctic has experienced dramatic reductions during the satellite era, with observed declines of –3.5% and –13.4% per decade in May and June, respectively (Meredith et al., 2019). Given the vast spatial scales and sparse in situ station network, remote sensing can play a critical role in snowpack monitoring in these sensitive environments.

The NASA SnowEx 2023 campaign was implemented to improve the understanding of remote sensing methods for retrievals of snow depth and snow water equivalent (SWE), the mass of the snowpack, in tundra and boreal forest environments (Vuyovich et al., 2024). These regions are understudied relative to temperate mountains; for instance, they are currently not included in gridded SWE analysis products (e.g., SNODAS). Space-borne remote sensing retrievals of snow depth or SWE at high spatial resolution may be achievable in these complex environments through lidar or radar methods (Fair et al., 2024; Eppler et al., 2022), but further evaluation of the uncertainties caused by the unique physical parameters within these environments (e.g., dense canopy cover and ground vegetation) is required.

The boreal forest is characterized by a dense canopy of coniferous and deciduous trees and covers 10–17% of the world's landmass (NASA Earth Observatory, 2006; Vuyovich et al., 2024). Boreal forest seasonal snowpacks are typically shallow (<1.2 m) and exhibit lower snow densities due to minimal wind loading, cold temperatures, and large temperature gradients in the snowpack, resulting in extensive faceting (Sturm and Liston, 2021). In contrast, the Arctic tundra extends to higher latitudes than boreal forests and is subject to extreme cold temperatures and high winds that limit plant growth. Tundra snowpacks are typically shallower than boreal forest snowpacks, but wind redistribution can build deep snow drifts (Sturm and Liston, 2021). Because of wind-driven compaction, tundra snowpacks can exhibit higher densities than snowpacks in the boreal forest and include a binary stratigraphy of high-density wind slab above low-density, large-grained depth hoar. Both environments have features that may complicate snow depth retrievals from lidar or radar methods. In boreal forests, dense canopy, canopy-intercepted snow, and dense shrub and tussock (i.e., small localized areas where the solid ground is raised) cover may occlude lidar, whereas dense shrub cover can cause a void space between the bottom of the snowpack and the ground, leading to a

Public domain. CC0 1.0.





mismatch between the more easily identified radar ground reflector and the true base of the snowpack. Tundra environments
contain spatially varying distributions of shrub and tussock cover that may increase uncertainty of lidar or radar methods.
Additionally, the North Slope tundra environments are underlain by shallow permafrost (Jorgenson et al., 2008) and exhibit
seasonal thaw cycles (e.g., thaw subsidence). This dynamic ground surface can cause increased lidar snow depth uncertainty
because of differences in the ground elevation between snow-on and snow-off lidar acquisitions (Chen et al., 2020). In both
environments, frozen soil can cause variable radar penetration below the snow-ground interface.
During NASA SnowEx 2023 in Alaska, we conducted detailed in situ surveys to improve our understanding of radar and
lidar performance for snow depth retrieval in these complex high-latitude environments. Here, we evaluate ground-based radar
and airborne lidar snow depth products along transects where manual measurements of snow depth were collected after snow
excavation. The excavated snow depths represent the integrated thickness of the snowpack, which excludes intra- and sub-
snowpack void spaces. In particular, we emphasize how sub-snow vegetation and variable ground conditions associated with
mosses, tussocks, and/or permafrost influence the retrieval accuracy.

## 2 Study Sites

The NASA SnowEx 2023 Alaska campaign (7–16 March 2023) was operated at three field sites in the boreal forests near
Fairbanks (Figure 1a–c) and two sites in the Arctic tundra on the North Slope (Figure 1d, e). Although some snowmelt was
observed in the boreal forest canopy, the snowpack on the ground was considered dry based on snow pit temperature
measurements. Of the boreal forest sites, the Caribou/Poker Creek Research Watershed site (CPCRW; Figure 1a) is located
~25 km northeast of Fairbanks and hosted eight surveys, the Farmers Loop Experimental Station and Creamer's Field
Migratory Waterfowl Refuge (FLCF; Figure 1b) north of Fairbanks hosted six surveys, and the Bonanza Creek Experimental
Forest (BCEF; Figure 1c) is ~20 km southwest of Fairbanks and hosted four surveys. Vegetation varied by site: CPCRW
surveys were performed primarily below black spruce canopy, FLCF included transects within black spruce (*Picea mariana*),
deciduous, mixed canopy, and agricultural field environments, and BCEF surveys were performed primarily below leaf-off
deciduous canopy and in wetland shrub environments (Vuyovich et al., 2024). Surveys at the Upper Kuparuk-Toolik site
(UKT; Figure 1d) had ground conditions that varied from shrubs and tussocks to frozen ponds, whereas the Arctic Coastal
Plain site (ACP; Figure 1e), located near Deadhorse, primarily included surveys performed in wetlands and frozen ponds/lakes
(Vuyovich et al., 2024). The sites near Fairbanks are within the discontinuous permafrost region (Jorgenson et al., 2008),
whereas both UKT and ACP are underlain by continuous permafrost (Obu et al., 2019).

Public domain. CC0 1.0.



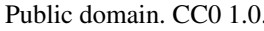

**Figure 1: Ground surveys at the (a) Caribou/Poker Creek Research Watershed (CPCRW), (b) Farmers Loop/Creamer's Field (FLCF), (c) Bonanza Creek Experimental Forest (BCEF), (d) Upper Kuparuk-Toolik (UKT), and (e) Arctic Coastal Plain (ACP) field sites. (f) Map inset depicts the locations of the five field sites within Alaska. Imagery provided by ESRI, Maxar and © Microsoft.**

## 3 Methods

### 3.1 Summary of Field and Airborne Lidar Surveys

We operated four different GPR systems across the various field sites. Details are provided in Text S1. For surface-coupled GPR surveys, the GPR was pulled across the surface of the snowpack in a sled, whereas air-coupled GPR was carried by two people above the snow surface. GPR systems were operated in the common-offset configuration and we took care to not disturb the snowpack immediately below the GPR transect. Transect lengths ranged from 3–12 m. Methods for deriving GPR snow depths are provided in Text S1. Following GPR data collection, the snow along the full length of the transect was excavated and snowpack thicknesses, which excluded any void spaces or tree branches/logs in the snowpack, were measured at 0.25–1 m intervals. Pictures and/or video recordings were acquired for each transect and detailed notes were taken on vegetation and ground conditions. An example photomosaic of an excavated boreal forest transect is provided in Fig. S1. Each survey was completed within 15 m of a snow pit, wherein measurements of snow depth, snow density, SWE, and snow temperatures were collected. In total, 17 surveys were collected at the boreal forest field sites and 27 surveys were collected at the tundra field sites on the North Slope. At each field site, snow-on airborne lidar surveys were collected during the campaign and snow-off airborne lidar surveys were collected during the following summer. Snow-on and snow-off lidar surveys were used to generate 0.5 m snow depth and canopy height models (Larsen, 2024). We note that lidar-derived snow depths based on acquisitions that

Public domain. CC0 1.0.

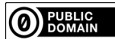



occurred before or after ground surveys have increased uncertainty due to either a changed snow surface (i.e., accumulation,
compaction, redistribution) between the lidar flights and ground surveys or snow disturbance caused by the ground
observations that preceded lidar flights (n = 20/44). Further processing details for the lidar snow depths are provided in Larsen
(2024). A list of lidar and ground survey dates is provided in Table 1 and snow accumulation measurements for every day of
the campaign are provided in Table S1.
**Table 1: Summary of field and airborne survey dates.**

|  | **Field Site** | **Snow-On Lidar Flight Dates** | **Ground Survey Dates** | **Transects (n)** |
|---|---|---|---|---|
| **Tundra** | **Arctic Coastal Plain** | 10 March 2023 | 11–14 March 2023 | 13 |
|  | **Upper Kuparuk-Toolik** | 13 March 2023 | 8–11, 15 March 2023 | 14 |
| **Boreal Forest** | **Farmers Loop/ Creamers Field** | 11 March 2023 | 7–11, 13 March 2023 | 8 |
|  | **Bonanza Creek Experimental Forest** | 11 March 2023 | 10, 13–15 March 2023 | 4 |
|  | **Caribou/Poker Creek Research Watershed** | 11 March 2023 | 8–9, 11, 14 March 2023 | 5 |


**3.2 Sources of Ground-Penetrating Radar Snow Depth Uncertainty**

For GPR systems, the transceiver emits a signal that transmits through the snowpack and reflects off boundaries of contrasting
dielectric permittivities (e.g., vegetation, snow stratigraphy, the snow-ground interface). The receiver then records the
amplitude and two-way travel times (*twtt*) of these reflections. Radargrams were processed and the *twtt* of the presumed snow-
ground interface were exported at ~0.10 m spacing, typically identified as the first reflection at depth with the highest amplitude
that is spatially coherent (further details provided in Text S1). Examples of two boreal forest radargrams are provided in Fig.
S2.
GPR surveys have several sources of uncertainty. GNSS positioning was estimated as ±0.5–3 m horizontal accuracy at
the boreal forest sites and ±0.5 m at the Arctic tundra sites, which could complicate the comparison between GPR and lidar
snow depths. GPR-derived snow depths may be reduced by up to 0.07 m due to removal and/or compaction of snow as the
sled travels across the surface. Finally, site-specific ground conditions may add uncertainty to the GPR snow depths. A few
examples include: (1) snow depths may be overestimated because of the presence of void spaces within the snowpack caused
by vegetation, (2) snow depths may be underestimated when a dense layer of vegetation obscures the true bottom of the

Public domain. CC0 1.0.

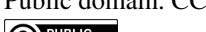



snowpack (e.g., a buried bent-over tree), and (3) snow depths may be overestimated if the radar signal penetrates into frozen
soil without a strong reflection at the snow-ground interface.

**3.3 Evaluation of GPR and Lidar Snow Depths**

We extracted the lidar snow depths (0.5 m x 0.5 m resolution) along each surveyed transect based on the GNSS coordinates
recorded by the GPR. We aligned the excavated depth locations with the GPR depth locations based on field notes, resulting
in a GPS uncertainty for the excavated depths that was identical to the GPR. GPR, lidar, and excavated depths were then
compared as individual profiles to evaluate any systematic differences. We calculated the mean and standard deviation of snow
depth for the GPR, lidar, and in-situ measurements from each transect to estimate the overall root mean squared error (RMSE),
bias, and Pearson's correlation coefficient (r) between measurement techniques in both the tundra and boreal forest sites
Finally, lidar-derived vegetation heights were extracted along each transect in the boreal forest to determine whether accuracy
was related to lidar-derived vegetation height.

**4 Results**

**4.1 Boreal Forest Transects**

At boreal forest sites, GPR mean snow depths overestimated excavated mean snow depths for 12 of 17 transects, whereas lidar
mean snow depths underestimated excavated mean snow depths for 14 of 17 transects (Table S2, Figure S3). Large differences
(>0.20 m) between GPR and excavated depths occurred where sub-snow shrubs or tree branches caused void spaces in the
snowpack (e.g., Figure S3g, i), but close agreement was observed for the transect in the agricultural field where only crop
stubble was present below the snowpack (Figure S3b) and for transects below black spruce canopy (e.g., Figure S3n–p). The
average residual between GPR and excavated depths was +0.06 m (9% of mean excavated depth) and the average residual
between lidar and excavated depths was –0.12 m (16% of overall mean excavated depth) when field surveys occurred after
lidar acquisitions (Table S2). The lidar snow depth bias worsened to –0.16 m (24% of mean excavated depth; Table S2) when
surveys performed before the lidar survey date were included in the comparison. We visually inspected the boreal forest lidar
snow depth rasters to verify the presence of all transects excavated before the 11 March lidar survey and we found that
excavation activities clearly influenced five of these surveys. These transects are shown in Fig. S3b, d, e, f, and n, illustrating
how sample timing can severely bias direct comparisons between field and remote sensing-based depth estimates.
Of the three boreal forest field sites, CPCRW yielded the lowest mean GPR residual (+0.03 m), whereas BCEF yielded
the highest mean residual (+0.11 m). Three of the five CPCRW transects were performed below black spruce canopy, where
mosses are the dominant substrate and shrub canopy is sparse, whereas shrubs and tussocks dominate the ground below BCEF
transects. FLCF surveys (mean GPR residual = +0.06 m) were performed primarily below mixed deciduous/coniferous canopy
and contained various shrubs, saplings, and tree branches within the snowpack. Compared to GPR, mean lidar residuals
exhibited the opposite canopy-driven trend; BCEF surveys yielded the lowest mean residuals (–0.10 m) and FLCF/CPCRW

Public domain. CC0 1.0.

yielded the highest mean residuals (–0.19 m and –0.15 m). However, we note that three of four BCEF surveys were performed
after lidar collection, whereas eight of 13 surveys at FLCF/CPCRW were performed before lidar collection (Table S2). Overall,
we calculated an r of 0.55 and RMSE of 0.14 m for GPR snow depths and an r of 0.51 and RMSE of 0.22 m for lidar snow
depths at the boreal forest field sites (Figure 2a–b). The best Pearson's correlation coefficient, but highest RMSE was observed
for the lidar vs. GPR comparison (r = 0.66, RMSE = 0.26 m; Figure 2c).

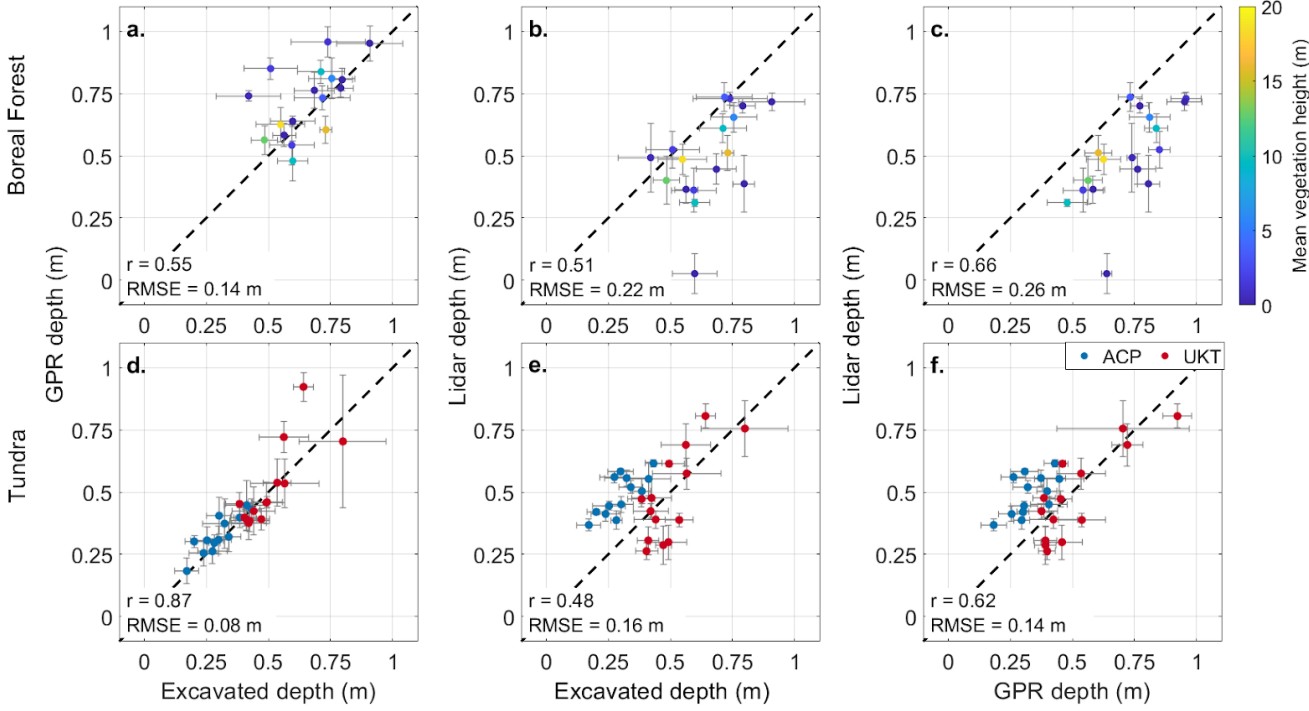


**Figure 2: Comparisons at the boreal forest (a–c) and tundra sites (d–f) between (a, d) GPR and excavated depths, (b, e) lidar and**
**excavated depths, and (c, f) lidar and GPR depths. The mean values are plotted with error bars calculated from the standard**
**deviation. Points in a–c are colored by mean lidar-derived vegetation height, which represents the canopy height in forest cover or**
**the shrub/grass height in meadows. Points in d–f are colored by field site. Sites: ACP, Arctic Coastal Plain; UKT, Upper Kuparuk-**
**Toolik.**

**4.2 Tundra Transects**
Vegetation structure was less complex at the Arctic tundra sites than at the boreal forest sites. At ACP, surveys had grassy
ground cover (11 surveys with grass heights of 0.05–0.20 m) or ground ice (two surveys; Figure S4i, m), whereas tussocks
(four surveys with tussock heights of 0.08–0.25 m) and shrubs (four surveys with shrub heights <0.30 m) were the predominant
ground cover at UKT. Based on the excavated depths, both mean GPR residuals (ACP = +0.03 ±0.04 m; UKT = +0.01 ±0.10
m; Tables S3–S4, Figures S4–S5) and the overall performance metrics (r = 0.87; RMSE = 0.08; Figure 2d) improved at the
Arctic tundra sites relative to the boreal forest sites.

Public domain. CC0 1.0.

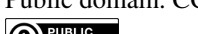



Compared to the boreal forest sites, the lidar snow depths from the Arctic tundra sites yielded a comparable Pearson's
correlation coefficient, but an improved RMSE (r = 0.48; RMSE = 0.16 m; Figure 2e). Despite having more complex ground
conditions than ACP, UKT yielded the lowest mean lidar residual (ACP = +0.19 ±0.05 m; UKT = –0.02 ±0.12 m; Tables S3–
S4, Figures S4–S5). Notably, all ACP surveys were performed after the lidar flight, whereas 13 of 14 UKT surveys preceded
the lidar flight (Table 1). We visually inspected the transects in the UKT lidar snow depth raster and observed that only five
transects were influenced by snow excavation (Figure S5h–j, l–m).
**5 Discussion**
**5.1 GPR and Lidar Snow Depth Uncertainty in Boreal Forest and Arctic Tundra Environments**
We found that GPR captured snow depth variability in both boreal forest and Arctic tundra environments over length scales
>0.5 m, but we observed larger uncertainty over shorter length scales (<0.5 m), likely due to the large sensor footprint (~1.5
m radius for a depth of 0.6 m; e.g., Daniels, 2004). GPR snow depths exhibited a modest positive bias (+0.06 m) relative to
excavated depths in the boreal forest and a smaller positive bias in the Arctic tundra (overall mean residuals = +1–3 cm; Tables
S2, S3). We attribute the positive biases to discontinuous vegetation-induced void spaces at the base and within the snowpack
(e.g., Figures S1, S2; Berezovskaya and Kane, 2007). Vegetation was dense and complex in the boreal forest, resulting in
cluttered radargrams that were difficult to interpret, whereas Arctic tundra sites had less complex vegetation and better defined
ground reflectors.
Lidar at ACP was expected to have the highest agreement with excavated depths because the lidar surveys were conducted
before the snowpack was disturbed. Instead, ACP lidar exhibited a +0.19 m average bias and failed to reproduce fine-scale
snow depth patterns (e.g., Figure S4a–b), despite having relatively simpler ground conditions at the time of the surveys.
Although UKT lidar exhibited lower mean bias, it also did not reproduce fine-scale snow depth patterns (e.g., Figure S5e–f).
Both sites received <0.06 m of snow depth accumulation between ground-based and airborne surveys (Table S1). We suggest
two primary sources for the larger lidar snow depth error at ACP: (1) blowing snow and wind redistribution (e.g., Pomeroy
and Li, 2000), which was observed on a daily basis, and (2) ground subsidence that occurred between the snow-on and snow-
off lidar survey, which was collected on 31 August 2022, near the end of the summer thaw cycle. Thaw has been observed to
drive subsidence up to 0.05 m in this region (Chen et al., 2020).
Lidar at the boreal forest also exhibited higher uncertainties, yielding an RMSE that was nearly half of the average
measured snow depth. Uncertainties tended to be higher for taller vegetation heights (>5 m; Figure 2b), although we note a
large range of accuracy for lower vegetation heights (<5 m). This weaker agreement could be due to one or more of four
primary factors: (1) snow disturbance due to ground surveys caused errors within the lidar retrieval method, (2) differences in
canopy structure between summer and winter acquisitions (e.g., bent-over trees from snow burden), (3) signal penetration
through the dense canopy may be problematic for the method (e.g., Hopkinson et al., 2004), or (4) GNSS location uncertainty

Public domain. CC0 1.0.

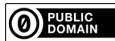



between the lidar and excavated depths. Despite the higher uncertainty for the lidar snow depths, the lidar coverage is spatially continuous and complete over the field sites, a clear advantage over the GPR measurements.

**5.2 Implications for SnowEx23 GPR and Lidar Datasets**

Although we identified a clear bias for GPR in the boreal forest, it appears that this bias exhibited some dependence upon the land cover, but we were unable to fully evaluate this point. Further work, potentially through a depth probe-GPR evaluation, is needed to identify systematic biases within the boreal forest for GPR operation.

At UKT, lidar snow depths exhibited low bias and relatively high accuracy, whereas lidar snow depths at ACP and the boreal forest sites exhibited a substantial positive bias. Fair et al. (2024) evaluated ICESat-2 lidar snow depths at UKT and FLCF using the airborne lidar snow depths and found better ICESat-2 performance at UKT ($r^2$ = 0.84–0.92) than at FLCF ($r^2$ = 0.14–0.58), further highlighting the boreal forest complexities. In the forests, the lidar snow depth bias is directly related to potential void spaces caused by shrub canopy interception and a lack of photon penetration below canopy and underbrush. In the Arctic tundra, lidar snow depths may be influenced by seasonal thaw cycles. Given the magnitude of snow disturbance associated with in situ surveys, we expect poor agreement between snow pit/excavated measurements and lidar measurements when the field survey preceded the lidar flight (e.g., Figure S3e, n).

**6 Conclusions**

Snow is a critical component of the hydrology and ecology of high-latitude terrestrial environments, but distributed snow depths are difficult to measure at high spatial resolution. As part of the NASA SnowEx March 2023 campaign, spatially distributed L-band GPR and airborne lidar data were collected and we evaluated snow depth retrievals at the boreal forest and Arctic tundra sites. Biases were observed in both environments, but ground-based GPR snow depths yielded higher overall accuracy. GPR, particularly when operated at or near the snowpack surface, is less sensitive to vegetation and snowpack void spaces than the airborne lidar DEM differencing method. We suggest additional complications related to the dynamic ground surface may arise for lidar, as exhibited by the large positive bias at ACP (bias = +0.19 m). The complex surface conditions of the boreal forest and tundra regions challenge established snow observing methods, warranting careful consideration when using these to evaluate novel remote sensing approaches.

**Author contributions**

Conceptualization: D.M., H.P.M., C.V., K.H.G., R.B., T.M., R.W. Data Curation: K.H.G., C.L., R.W., T.M., M.M., R.B., D.M., B.B. Analysis: K.H.G., R.B., T.M., R.W. Funding Acquisition: D.M., C.V., H.P.M., R.W., R.B., B.B., T.M., K.H.G.



Investigation: all authors. Methodology: K.H.G., D.M., T.M., R.W., R.B. Visualization: K.H.G, R.B. Writing – Original Draft
Preparation: K.H.G, R.B., D.M. Writing – Review & Editing: all authors.

**Acknowledgements**

K.H.G was supported by the Colorado State University Honors program. R.B. was supported by NASA FINESST award
80NSSC20K1624 and the U.S. Geological Survey Mendenhall Postdoctoral Fellowship Program. D.M., H.P.M., and R.W.
were supported by NASA THP award 80NSSC22K1113. We thank Dr. E. Baker, H. Flynn, and N. Latysh for providing
insightful comments which improved the clarity of this paper. Any use of trade, firm, or product names is for descriptive
purposes only and does not imply endorsement by the U.S. Government.

**Data Availability**

GPR transects (Bonnell and McGrath, 2024; Meehan and Rowland, 2024; Webb, 2024), lidar data (Larsen, 2024), and snow
pits (Mason et al., 2024) are archived with the NSIDC DAAC.

**Competing Interests**

At least one of the (co-) authors is a member of the editorial board of *The Cryosphere*. The authors have no other competing
interests to declare

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
