# Peer review of "Brief Communication: Evaluating Snow Depth Measurements from"

_EGUsphere, 2025_

## Author Comment (AC1)

**Response to Reviewer 1**

**Dear Dr. Matthew Sturm,**

Thank you for your thorough review and constructive feedback on our paper. Given your comments regarding the supplemental figures, we have restructured our manuscript as a Research Article to alleviate figure and citation constraints. Below, you will find responses to each of your points in blue.

Thank you very much for your time and thoughtful comments,

Kajsa Holland-Goon and Randall Bonnell, on behalf of co-authors

Review of: Evaluating Snow Depth Measurements from Ground-Penetrating Radar and Airborne Lidar in Boreal Forest and Tundra Environments during the NASA SnowEx 2023 Campaign

This is a nice tidy paper with useful data, but I think it reaches conclusions that a more thorough analysis might contradict. I spent a while studying figures S3, S4 and S5, the real heart of the paper, and think it would improve the paper if the authors went back to these figures and spent a bit more time thinking about them and how the various data traces relate to each other.

First off, Figures S3, S4, and S5 are the *real* results and ought to appear in the paper itself, not just as supplemental material. The statistical summaries in Figure 2 are useful, but they don't allow a reader to examine the data in detail. Moreover, the data in Figure 2 are from the first two moments in statistics (mean and variance), but these do not make use of the spatial organization of the snow depth data (e.g., the relative relationships of ups and downs in a depth transect). Figures S3, S4 and S5 do that, and provide a reader with better sense of how well the remote sensing sampling (GPR and lidar) reproduced the depth point data profile. I would highly recommend to the authors reading a paper by Blöschl and Sivapalan (1995) that discusses *support*, *extent* and *spacing* in snow sampling. In this study, where those differ so much, it might help in framing the comparison and conclusions.

Blöschl, Günter, and Murugesu Sivapalan. "Scale issues in hydrological modelling: a review." *Hydrological processes* 9, no. 3-4 (1995): 251-290.

Here are a few points that are not addressed in the paper, but could be important:

1. The GPR surveys were done at very small scale (a few meters), so despite the GPR's larger areal footprint, we might expect the point-to-point co-registration between the GPR data and the excavated snow depths to be excellent. The co-registration between the airborne lidar and excavated depths is done by GPS, and has stated potential error of 50% of the transect length. The lidar requires two co-registrations: between snow-on and snow-off acquisitions (which gets done months later), and between that result and the excavated depths. The lidar aircraft flies at least 2000'away, and all its depths, and position of the depths, relies on a black-box solution using Metashape or Pix4D. I don't think co-registration of lidar to the depth transects can be assumed, and perhaps it is prudent to assume it is likely to be off.

Yes, we expect high quality co-registration between the GPR and the excavated depths. For each transect, we first marked out the transect before the GPR survey was conducted. After the GPR survey, we used a tape measure to mark every 0.25 m along the excavated trench for a snow depth measurement. Then, we dug out the snow from below the transect and conducted our in situ measurements with a ruler. We have added a sentence to section 3.3:

"We aligned the excavated depth locations with the GPR depth locations based on field measurements, resulting in high quality co-registration between the GPR and the excavated depths. However, the co-registration between the lidar and the transect surveys is subject to the GPS uncertainty listed in Section 3.2."

One point of clarification is that the lidar point clouds are generated from an established and well-documented approach

(https://nsidc.org/sites/default/files/documents/user-guide/snex23\_lidar-v001-usergui de\_2.pdf) , rather than a black-box solution. Both Metashape and Pix4D software programs are commonly used for Structure-from-Motion processing of optical imagery, but were not used in the lidar processing. As detailed in the User Guide linked above, the SnowEx lidar datasets start with aircraft trajectory solutions from data recorded by an Applanix POS AP60-AV Inertial Navigation unit. These trajectory solutions are well-located and lead to point clouds that are georeferenced with high enough accuracy that no adjustments are needed when co-registering to create difference maps. All SnowEx lidar datasets align within ±0.05 m horizontally, and all but the ACP lidar dataset align within ±0.05 m vertically. We elaborate on identified issues at ACP in our response to point 6. Below, we illustrate the horizontal alignment between GPR/excavated depths and lidar data.

Figure: Alignment of lidar at GPR and excavated depth transects in boreal forest sites. Selected transects were performed before the lidar survey date and the excavated trenches are evidenced by nearly 0 m snow depths in each of the plots. Transects in plots e and f were collected with the 1.6 GHz GSSI GPR system, which had a GPS uncertainty of ±3 m. The remaining transects were collected with the 1.0 GHz Sensors & Software GPR system, which had a GPS uncertainty of ±0.5 m.

In the above figure, plots a, c, d, and f show excellent agreement between the GNSS solutions for lidar and the transects. Plots b and e show an offset between the GPR survey and the lidar excavated pit. The transects in plots b and e are within 2 m and 0.5 m of the trench footprints, respectively. We agree that lateral offsets may cause higher residuals in the lidar vs. excavated depth comparison. Thus, we have emphasized point (4) in Section 5.1, that GNSS uncertainty may be a factor driving the residuals in the lidar snow depths.

2. The authors ascribe the offsets between the GPR and ruler depths to air gaps, but say nothing about what happens when one compares depths averaged over a substantial circular area (over a bumpy substrate of tussocks) to a two-dimensional profile in which the edges of 3D bumps appear as a simple 2D curve. This *support*

(Blöschl and Sivapalan, 1995) mismatch could easily explain some or even a lot of the differences between the GPR and the ruler depths. Ironically, though the lidar is run thousands of feet away from the target, its support has the putative dimensions of a photon (or there-abouts) and so theoretically matches the excavated depth support better.

Thank you for sharing these astute observations and associated reference. GPR does indeed have a much different support than either lidar or ruler depths. The GPR footprint is approximated by a Fresnel cone that can be calculated from the radar signal wavelength and the distance from the source antenna to the reflector of interest. The cone expresses itself as an approximately circular area, depending on the slope and complexity of the terrain and the returned radar signal is effectively an averaged travel-time over the circular area. The circular area is many times larger than the ruler depth support. We have revised Section 5.1 to include these details.

3. Which brings me to what I call the differences in work flow of the two remote sensing methods (e.g., the processing that has to happen with these two types of remote sensing efforts to produce a depth value.) For the GPR, a human picks the ground surface, a little subjective, but an expert process, and done for a very limited area. For the lidar, that happens in a black box, requires two flights separated in time, and any co-registration mismatch between the first and second acquisitions will produce errors (not biases). It is done for a huge area all at once. So I would question how relevant it is to test airborne lidar data that can cover many square kilometers using such limited scale (5- to 10-m) ground trench data.

Yes, it is challenging to evaluate broader scale remote sensing products with the much smaller-in-scope data collected during ground campaigns. The snow depth probe data would offer a much larger dataset for evaluation than our limited ground transects. However, leading up to the SnowEx Alaska campaign, there was much discussion regarding potential uncertainties of traditional snow depth probing in these environments. Part of the science plan (Vuyovich et al., 2024) was to evaluate the traditional snow depth probe measurements with ruler depths obtained in excavated holes directly below the snow depth transects. This process was very similar to our methodological approach for the excavated trenches. Recently, the snow depth magnaprobe measurements were evaluated against the detailed excavated depths and a bias of +0.10 m was observed for the snow depth probe measurements at the boreal forest sites (see figure below from Stuefer et al., 2025, https://doi.org/10.1038/s41597-025-05170-x).

Figure: Evaluation of traditional depth probing using detailed excavated snow depths directly below the depth probe transects. From Stuefer et al. (2025).

We argue that this paper's findings support the use of the excavated snow depths for evaluating the collected datasets, whether at the smaller scale (e.g., GPR) or the much broader scale of the lidar. The excavated depths are likely the most detailed and most reliable ground-based dataset collected by the SnowEx team.

As we described earlier, lidar elevations are derived from a well-established and documented workflow. We agree that the point about errors vs. biases is important to address and we have revised Section 3.3 to include this nuance. We have decided to keep the results and discussion focused on biases, rather than errors, because errors are difficult to disentangle from biases in these complex environments. We discuss this in further detail for the fourth major point and we note that ACP is an exception, as outlined in our reply to point 6.

4. That said, looking at Figures S3 to S5, the lidar profiles seem to occur in two modes for the taiga areas: either the lidar falls right on top of the excavated depth profiles (FLCF 11 March DN091) or it is way off (FLCF 9 March WB032), but in many cases, parallel to the excavated depth profile. This binary behavior suggests to me that in the lidar data we are seeing both *biases* and *errors*. The biases are where something like frost heave or void spaces have produced differences between the lidar and the probe depth; the errors are where the co-registration or the GPS bundle solution, or something else, has gone wrong.

Yes, as noted, there appear to be three unique patterns for the lidar profiles. Here, we focus exclusively on transect data collected *after* the lidar survey date:

(1) Good agreement spatially, good magnitude agreement: Boreal Forest Transects:

11 Mar DN091 (Figure 3g) – With the exception of the decreased snow depth in the middle of the transect, the lidar profile nicely follows and agrees with the excavated profile.

13 Mar DB337 (Figure 3j) – The lidar profile nicely follows and agrees with the excavated profile, with a slight underestimation of 0.06 m.

11 Mar WA282 (Figure 3p) – Limited overall variability for both lidar and excavated depths, but lidar patterns agree with excavated patterns and exhibit minimal bias.

14 Mar EA229 – (Figure 3q) – lidar misses some of the variability along the transect, but, for the most part, produces a representative and accurate depth profile.

**Upper Kuparuk-Toolik Transects**

15 Mar D698 (Figure 6n) – with the exception of the middle portion of the transect, the lidar profile nicely follows the excavated profile with nearly matching agreement.

**(2) Good spatial pattern agreement, poor magnitude agreement:**

**Arctic Coastal Plain Transects:**

12 Mar A500 (Figure 5c) – Lidar profile nicely follows excavated profile, but overestimates depth by an average of 0.14 m.

13 Mar A522 (Figure 5f) – For the most part, the lidar profile follows the excavated profile, but overestimates depth by an average of 0.19 m.

13 Mar I529 (Figure 5i) – the lidar profile mimics the excavated profile, but misses some of the variability. Lidar overestimates depth by 0.17 m.

14 Mar I549 (Figure 5m) – Lidar profile nicely follows the excavated profile, overestimates depth by 0.11 m.

**(3) Poor spatial pattern agreement, poor magnitude agreement:**

**Boreal Forest Transects:**

13 Mar DB106 (Figure 3h) – lidar has very little variability and underestimates snow depth.

14 Mar SA326 (Figure 3k) – the lidar profile has little variability compared to the complex variability illustrated by the excavated profile. The lidar underestimates snow depth by 0.11 m.

15 Mar WA437 (Figure 3I) – There is a lot of variability in the excavated depths, but the lidar depths are effectively a flat line.

11 Mar WN281 (Figure 3o) – Low variability for the lidar profile, lidar underestimates excavated depths.

**Arctic Coastal Plain Transects:**

All except the four listed above show poor spatial agreement and poor snow depth value agreement. For six of the transects, the lidar profile is represented by a line with almost no variability.

To summarize, we see depth profiles distributed across all three of the above categories for transect data collected after the lidar surveys. Notably, the Arctic Coastal Plain lidar data falls nearly exclusively into the second two categories (good spatial agreement/poor magnitude agreement and poor spatial agreement/poor magnitude agreement). Neither the boreal forest transects nor the Upper Kuparuk-Toolik transects fall within the second category. We have added these notes to Sections 4.1 and 4.2.

In response to major comment 6, we investigate the potential causes of the Arctic Coastal Plain lidar residuals.

5. The Kuparuk-Toolik results also seem to show a similar behavior: there are some nice lidar to depth matches but also some poor ones: for example, 9 Mar N789 is "on" for both the GPR and the lidar, but for 11Mar A739 the lidar has a distinct low bias across the whole transect. That stands as an argument that substrate character alone is not necessarily the factor driving the bias in the lidar.

Lidar results from the Upper Kuparuk-Toolik site are challenging to interpret, primarily because nearly all transects were dug before the lidar flight. Field notes from this site indicate nearly constant wind redistribution throughout the campaign. Boreal forest trenches dug before the lidar flight have a distinct appearance in the lidar snow depths (as shown above), whereas the transects in the UKT lidar are nearly indiscernible compared to the surrounding lidar snow depths (see figure below). We suspect that UKT trenches were filled in with redistributed snow before the lidar flight. In most cases, the average lidar snow depth value compares nicely with the excavated value, but, as we noted in the Section 5.1, UKT often fails to capture the spatial heterogeneity observed in the excavated depths.

Figure: GPR/excavated depth surveys overlain on the 13 March 2023 UKT lidar survey. In most cases, the excavated areas are indistinguishable from the surrounding snow depth patterns (e.g., panels a and b). However, for transects conducted on 11 March, some patterns of the field activities can be discerned. For example, the snow depth spiral

(collected as part of the SnowEx survey plans) and part of the trench can be discerned in panel c.

6. Which brings me to the puzzling results from the Arctic Coastal Plain (ACP). I have worked extensively with Chris Larsen collecting airborne lidar across the ACP, and I have validated those results with literally thousands of on-the-ground snow depths (see Sturm et al., 2019 (this report to BLM and the USFWS is hard to find now, so I attach it); see also Nolan et al., 2015). It has been our experience in this work that often there is an affine translation in the lidar raster depth surveys that needs to be corrected to produce a useful survey. Basically, due vagaries in the 3D GPS solution, or perhaps it is minor issues with the GPS constellation, the entire survey may float too high or drop too low. Anyway, that affine translation needs to be corrected by using ground validation data. It doesn't take much ground data to do so. If there are no in situ snow measurements, we have used snow-free patches to do this. So the lidar data for the ACP 13 Mar A522, adjusted downward about 12 cm, gives a fine match to the excavated depth data, suggesting that some of the lidar data have issues like this. Apropos to the SnowEx goal of validating an airborne radar retrieval, I would simply take the extensive magnaprobe field data from SnowEx, and "calibrate" the lidar flight data against it to produce a B-level corrected product, then use that for comparison to the radar results.

After further investigation, it appears that the ACP lidar snow depth dataset has vertical alignment issues along the swaths that were not easily identified during our earlier analysis. To illustrate the issue, we provide a figure below, where the median snow depth from the ACP lidar snow depths dataset (0.46 m) is subtracted from the snow depths, resulting in an anomaly figure. These lidar artifacts are now obvious as linear "striping" in a northeast to southwest direction. Unfortunately, most field surveys were coincidently conducted within a single swath exhibiting a deeper than average lidar snow depth. We thus conclude that these artifacts within the ACP lidar snow depths caused the observed bias compared to our excavated depths. We have added these details and the figure to the discussion in Section 5.1.

Figure: Lidar snow depth anomaly at ACP. Linear striping is a result of the lidar survey design, wherein no cross-track surveys were conducted.

We reviewed the lidar survey designs at each site. ACP is the only site without any cross-track surveys to constrain vertical alignment between the swaths. We then checked the remaining lidar snow depth rasters from the other four field sites. We did not identify any evidence of swath vertical alignment issues within the boreal forest sites, but did note that the northeastern arm of the UKT lidar snow depths raster exhibits minor striping in a north-south orientation. However, this minor striping aligns directly with topographic features and thus may reflect true snow depth patterns rather than an artifact of the lidar processing.

Nolan, M., Larsen, C., & Sturm, M. (2015). Mapping snow depth from manned aircraft on landscape scales at centimeter resolution using structure-from-motion photogrammetry. *The Cryosphere*, *9*(4), 1445-1463.

Summary: In the end, I felt like this paper was comparing apples to oranges, or perhaps cherries (GPR) to watermelons (lidar), and of course, finding differences. The lidar is designed to cover hundreds of square kilometers, but at the cost of a complex technical

solution that can show accuracy drift and needs to be "calibrated" in practical use. The GPR is good at providing a detailed solution on a small patch of snow, using a human interface for handling the complexity of "picking" the base of the snow. It imposes strong spatial averaging. The data presented are interesting data, but I think the explanations for the differences in remote sensing methods presented in the paper are neither fully correct nor nuanced enough. I suggest going back after reading Blöschl and Sivapalan (1995) and looking at those supplemental figures more closely.

We thank Dr. Sturm for his careful review of our manuscript. Without his insight, we may not have identified the anomalies in the ACP dataset.

**Minor Points:**

Lines 26-28: Snow also strongly impacts the ecology of these regions: snow influences caribou winter range selection (Pedersen et al., 2021) and vegetation phenology (Kelsey et al., 2021), and provides winter refuges for a diverse range of animals (Penczykowski et al., 2017).

This is a point I try to make (usually unsuccessfully) to young investigators: it is silly to reference a 2017 or 2021 reference to make general points about snow in the North. There are so many earlier papers that established that point...some dating back 70 years or more. To fail to credit all that great older seminal work is to suggest it didn't happen. I think I would prefer no citations to buttress the statement in the text than a cursory sprinkling in of some newer citations that suggest the old work never happened. Similarly, where the authors cited my work (Sturm and Liston, 2021) as evidence for wind slabs on the tundra and faceted grains in the taiga, I cringed. That 2021 paper is global in scope; to lead a reader to a useful reference on depth hoar of wind slab, how about citing my mentor, Carl Benson (now 98):

Benson, C.S. 1967: Polar Regions Snow Cover, In Physics of Snow and Ice: Proceedings, 1(2), 1039-1063. *International Conference on Low Temperature Science. I. Conference on Physics of Snow and Ice, II.* Conference on Cryobiology. (August, 14-19, 1966, Sapporo, Japan),

**or if you must:**

Benson, C. S., & Sturm, M. (1993). Structure and wind transport of seasonal snow on the Arctic slope of Alaska. *Annals of Glaciology*, *18*, 261-267.

Because we submitted as a Brief Communication, we were limited to 20 references. We have revised our manuscript into a Research Article and have added several citations to further support our introductory statements:

Duquette (1988) - Snow characteristics along caribou trails and within feeding areas during spring migration

Aitchison (1987) - Winter energy requirements of soricine shrews

Pruitt (1970) - Some ecological aspects of snow

Benson (1967) - Polar Regions Snow Cover

Benson and Sturm (1993) - Structure and wind transport of seasonal snow on the Arctic slope of Alaska

However, we have also elected to keep the previously cited papers as references because these papers studied relevant components of the Arctic tundra and boreal forest environments.

Line 74: There are tussocks, and there are hummocks, and there are ice wedges and polygons. All combine to make the tundra a bumpy surface. Perhaps be a bit more general here.

Here, we summarized the field notes taken by SnowEx field participants (including ourselves) in the excavated trenches and in the snow pits. These notes did not contain any details about hummocks, ice wedges, and ice polygons, so we do not think that the inclusion of these features at Line 74 is appropriate. However, we agree that we need a few more details on these features and have revised lines 52–54 in the Introduction (Section 1) to better describe the ground surface of the Arctic tundra.

Line 141: Can we assume that after the GPR pass there were foot holes and sled marks in the snow, and that the excavation resulted in both a trench and heaped up pile of snow behind the trench? So there were many square meters of messed up snow. Perhaps any lidar that was done post-excavation ought to be culled from the paper.

Yes, we can assume that the GPR and excavation activities left a dug out trench and a pile of snow. We provide a figure illustrating this issue in our response to point 1. This is why we report statistics for both the ground surveys conducted *after* the lidar survey and the full dataset. Many of our field activities significantly impacted the snow cover in the lidar datasets and we feel it is important to highlight this complexity of campaign data collection.

Line 155-Figure 2: Pretty clear that the ACP and UKT data differ in some fundamental way for the lidar. This then drives the difference between lidar and GPR for the tundra. I think it might be more useful when presenting the taiga results to color-code the data by site rather than canopy/vegetation height. Then we could see if there is a site bias in the lidar for the taiga as well as the tundra.

In Section 4.1, we reported the residuals as a function of vegetation type:

- Transects below mature black spruce were primarily performed at Caribou/Poker Creek Research Watershed. We conjectured that the relatively simple ground conditions below the black spruce were ideal for GPR, but that the dense canopy and the canopy-intercepted snow occluded the lidar signal.
- 2. Transects in the open, where tussocks and shrubs were the dominant ground cover, yielded the worst results for GPR, but the best results for lidar. These transects were primarily collected at Bonanza Creek Experimental Forest. Here, radar reflections from the dense sub-snowpack/intra-snowpack vegetation cluttered the radargrams, whilst the lidar sensor likely had a clear view of the snow surface.
- 3. Transects in mixed canopy or deciduous canopy yielded mixed results. These transects were primarily surveyed at the Farmers Loop/Creamers Field site. In some cases, many young trees comprised the ground conditions and caused increased uncertainty for the GPR. In other cases, ground conditions simply consisted of leaf litter and the GPR performed very well.

Thus, we feel that we have represented the data accurately in the figure and we would recommend that readers consult the supplemental tables to identify the site-based biases at the boreal forest.

Line 217-219: Conclusions: Sorry, but I am just not convinced this statement is valid. The data indeed differed between methods, but of [course] there are so many variables affecting the results of each remote sensing method that citing just void spaces and vague reference to vegetation effects gives the wrong impression. Don't ignore the vast differences in support, spatial sampling between the methods, as well as the work-flow differences in the data reduction.

We have revised these lines to: "Biases were observed for both methods in both environments. GPR surveys yielded accurate snow depths, but with lower variability along the profile, which was likely caused by the much larger footprint of the radar signal. GPR, particularly when operated at or near the snowpack surface, is less sensitive to vegetation and snowpack void spaces than the airborne lidar DEM differencing method. The timing of the lidar flights relative to the ground surveys complicated our evaluation, but we noted reduced accuracy in the boreal forest that may have been caused by vegetation occluding the lidar signal. Although lidar yielded accurate snow depths at the UKT site, we identified vertical alignment issues that caused a large positive bias at ACP (bias = +0.19 m)."

**References**

Aitchison, C.W. (1987). Winter energy requirements of soricine shrews. *Mammal Review*, 17(1), 25-38. https://doi.org/10.1111/j.1365-2907.1987.tb00046.x

Benson, C.S. (1967). Polar Regions Snow Cover. *Physics of Snow and Ice: proceedings*, 1(2), 1039-1063.

https://eprints.lib.hokudai.ac.jp/dspace/bitstream/2115/20360/1/2\_p1039-1063.pdf

Benson, C.S., and Sturm, M. (1993). Structure and wind transport of seasonal snow on the Arctic slope of Alaska. *Annals of Glaciology*, *18*, 261-267. https://doi.org/10.3189/S0260305500011629

Duquette, L.S. (1988). Snow characteristics along caribou trails and within feeding areas during spring migration. *Arctic*, *41*(2), 143-144. https://doi.org/10.14430/arctic1706

Pruitt, W.O. (1970). Some ecological aspects of snow. *Ecology of the subarctic regions:* proceedings of the Helsinki Symposium, pg. 83–99. https://unesdoc.unesco.org/ark:/48223/pf0000004082

Stuefer, S.L., Hale, K., May, L.D., Mason, M., Vuyovich, C., et al. (2025). Snow depth measurements from Arctic tundra and boreal forest collected during NASA SnowEx Alaska campaign. *Scientific Data*, *12*(1), 919. https://doi.org/10.1038/s41597-025-05170-x

Vuyovich, C., Stuefer, S., Gleason, K., Durand, M., Marshall, H.P., et al. (2024). NASA SnowEx 2023 Experiment Plan. Science Plan. https://snow.nasa.gov/sites/default/files/users/user354/SNEX-Campaigns/2023/NASA\_SnowEx Experiment Plan 2023 draft 20June2024.pdf

---

## Author Comment (AC2)

**Response to Reviewer 2**

**Dear Dr. Andrea Vergnano,**

Thank you for your constructive and positive comments! We found your suggestions regarding the difficulties of flipping between the supplement and main manuscript to be helpful, and have restructured our manuscript accordingly. Below, we provide detailed responses to each of your points (our responses are in blue).

Thank you very much for your time and expertise,

Kajsa Holland-Goon and Randall Bonnell, on behalf of co-authors

Dear authors, I had the opportunity to review your manuscript entitled "Brief Communication: Evaluating Snow Depth Measurements from Ground-Penetrating Radar and Airborne Lidar in Boreal Forest and Tundra Environments during the NASA SnowEx 2023 Campaign".

**General comments:**

In your work, you assess the lidar accuracy to map snowpacks in high-latitude environments, with a focus on boreal forests and tundra environments. You perform a comparison with GPR and manual excavation in several transects. The manuscript is clear and well-written, and highlights the importance of assessing instrument uncertainties in mapping the snow accumulation.

I am not a lidar expert; therefore, I do not comment on it. However, I performed GPR measurements in snowpacks. The GPR data collection, the instruments used, and the resulting radargrams are of good quality.

I appreciated your work because you highlighted the possible causes of the observed uncertainties, which I find very useful for further research. You do not always investigate in detail such causes, which you leave for future research, but I think that with the data you have, you could already extract more detailed correlations. Moreover, sometimes I found it a little difficult to follow your text, because the figures are in the supplementary materials, and I think that your manuscript lacks a figure in which you show the GPR radargram, the photomosaic and the lidar depth together on the same transect.

I do not find severe problems in the manuscript, but it may be improved if the relation between instrument uncertainties and their causes is discussed in more detail. I suggest the manuscript to be accepted after minor revisions.

Thank you for your review and suggestions. We particularly appreciated your review of the GPR component of our paper. We agree that flipping between the supplement and the main manuscript was frustrating and we have restructured our manuscript as a Research Article, instead of a Brief Communication. As such, we have moved supplemental figures and text to the main text, but propose keeping the tables in the supplement as those elements provide supporting information to the primary findings. Additionally, we have decided to revise Figures S1 and S2 to a single figure (now, figure 2 in the main text) that connects the physical features shown in the photomosaics to the associated reflections in the radargrams.

**Specific comments:**

Figure 1: Please, add a scalebar to panel f). Additionally, consider adding the location of Fairbanks in panel f), since it was mentioned several times in the text.

We have revised the figure and it is pasted below. Thank you for catching it.

Figure: The updated field site figure for the main manuscript. Changes were made to panel f - we added a scale bar and the location of Fairbanks.

Data availability. Please, add a link to the NSIDC DAAC repository. Also, the fact that you put a part of data availability in the main text and a part in the supplementary material is confusing, in my opinion.

We have added links to each of the datasets listed in the Data Availability section and we have updated the availability section such that the supplemental materials match the main manuscript. Thank you for catching these issues.

Chapter S1 Ground-Penetrating Radar Systems and Methods: consider adding more details about which instrument was used in which site. Especially, one of the instruments (the GSSI one) had worse GPS positioning than the others. It would be important to assess if this introduced greater uncertainties in the GPR-lidar comparison.

Figures 1 and S3–S5 (now, figures 3, 5, and 6) note what instruments were used at each site. Additionally, we have added GPR system information to supplemental tables S2–S4.

Of the boreal forest transect surveys performed *after* the lidar surveys, three were collected with the 1.0 GHz 1 polarization (Sensors & Software) system, whereas five were collected with the 1.6 GHz 1 polarization (GSSI) system. The GSSI system was noted to have a larger GPS uncertainty, yet the GSSI surveys yielded lower biases for the lidar depth vs. excavated depth (median bias = -0.09 m) when compared to the Sensors & Software system (median bias = -0.22 m). We have added this note to Section 5.1. Thank you for the suggestion.

Figure S1: I think that you missed a great opportunity to show the DN013 radargram (and the lidar estimated depth) here. If you do so, you could elaborate on the buried vegetation that contributed to the GPR and lidar uncertainties (e.g. "at transect distance = 5 m, a buried little tree is also shown in the radargram as these hyperbola circled in red in the radargram, and this had this ... negative effect on the lidar snow depth estimation").

We think this is a great idea and we are working on a figure to add as Figure 2 to the main text. Thank you for the suggestion and we hope this revision improves readers' understanding of our research.

Figure S2: I am not convinced that the resolution difference you mention in the Figure caption is due to the different antenna frequency. In my experience, 1 GHz is already enough to show features as little as those recorded by the 1.6 GHz antenna, maybe just a little worse. I suppose that the perceived resolution difference between the two images is just due to the different spatial resolution: DB254 seems to have 1 trace per 10 cm, while SA326 has a much higher spatial resolution (I can't count the pixels, but they are much more than DB254). Also, similar to what I told for Figure S1, I really would like to see if the snowpack photomosaic of DB254 is different from that of SA326, to more constructively assess if the differences in the radargrams are related to the vegetation buried under the snowpack.

Upon evaluating this comment, we found an issue with figure S2 - the x-axes were not scaled the same, which gave the appearance of higher horizontal resolution for the 1.6 GHz

antenna. The newly updated Figure 2 will show a matching radargram and photomosaic. We agree that our comments regarding the vertical resolution of the GPR may be misleading and have removed these statements.

Figures S3 to S5: I would put at least one of them in the main text. Also, in the legend, please note on which date the lidar survey was performed, so it is easy to visually assess which transects were surveyed before and after the lidar.

Following your previous comments regarding readability challenges, we have decided to add the supplemental figures and supplemental text to the main text and reformat our article as a Research Article. Thus, Figures S3–S5 (now figures 3, 5–6) have been moved to the main text and we have added a note on the lidar survey date to the caption of each figure to improve the interpretability. Thank you for these suggestions.

Supplementary text, line 126: a full stop is missing at the end of the line. Thank you for catching this mistake.